# Transformer-based deep learning enables improved B-cell epitope prediction in parasitic pathogens: A proof-of-concept study on *Fasciola hepatica*

Rui-Si Hu[1,2], Kui Gu[3]*, Muhammad Ehsan[4], Sayed Haidar Abbas Raza[5], Chun-Ren Wang[6]*

1 School of Health and Wellness Industry & School of Medicine, Sichuan University of Arts and Science, Dazhou, Sichuan Province, People's Republic of China, 2 Key Laboratory of Intelligent Medicine and Health Data Science, Sichuan University of Arts and Science, Dazhou, Sichuan Province, People's Republic of China, 3 School of Pharmacy and Bioengineering, Chongqing University of Technology, Banan, Chongqing, People's Republic of China, 4 Department of Parasitology, Faculty of Veterinary and Animal Sciences, The Islamia University of Bahawalpur, Punjab, Pakistan, 5 State Key Laboratory of Biocontrol, School of Life Sciences, Sun Yat-Sen University, Guangzhou, Guangdong Province, People's Republic of China, 6 College of Animal Science and Veterinary Medicine, Heilongjiang Bayi Agricultural University, Daqing, Heilongjiang Province, People's Republic of China

* gukui0404@cqut.edu.cn (KG) chunrenwang@sohu.com (CRW)

## Abstract

### Background

The identification of B-cell epitopes (BCEs) is fundamental to advancing epitope-based vaccine design, therapeutic antibody development, and diagnostics, such as in neglected tropical diseases caused by parasitic pathogens. However, the structural complexity of parasite antigens and the high cost of experimental validation present certain challenges. Advances in Artificial Intelligence (AI)-driven protein engineering, particularly through machine learning and deep learning, offer efficient solutions to enhance prediction accuracy and reduce experimental costs.

### Methodology/Principal findings

Here, we present deepBCE-Parasite, a Transformer-based deep learning model designed to predict linear BCEs from peptide sequences. By leveraging a state-of-the-art self-attention mechanism, the model achieved remarkable predictive performance, achieving an accuracy of approximately 81% and an AUC of 0.90 in both 10-fold cross-validation and independent testing. Comparative analyses against 12 handcrafted features and four conventional machine learning algorithms (GNB, SVM, RF, and LGBM) highlighted the superior predictive power of the model. As a case study, deepBCE-Parasite predicted eight BCEs from the leucine aminopeptidase (LAP) protein in *Fasciola hepatica* proteomic data. Dot-blot immunoassays confirmed

**Data availability statement:** The MaxQuant search results, based on entries from the UniProt database, can be downloaded from the Zenodo repository at https://doi.org/10.5281/zenodo.14065139. The source code for the deepBCE-Parasite software is available on GitHub at https://github.com/RuiSiHu/deepBCE-Parasite and is also archived in the Zenodo repository at https://doi.org/10.5281/zenodo.14907455.

**Funding:** This work was supported by the National Natural Science Foundation of China (Grant No. 62202083 to RSH), the China Postdoctoral Science Foundation (Grant No. 2022M710618 to RSH), and the High-level Talent Research Start-up Project of Sichuan University of Arts and Science (Grant No. 2024GCC16Z to RSH). The funders had no role in study design, data collection, and analysis, decision to publish, or preparation of the manuscript.

**Competing interests:** The authors have declared that no competing interests exist.

the specific binding of seven synthetic peptides to positive sera, validating their IgG reactivity and demonstrating the model's efficacy in BCE prediction.

## Conclusions/Significance

deepBCE-Parasite demonstrates excellent performance in predicting BCEs across diverse parasitic pathogens, offering a valuable tool for advancing the design of epitope-based vaccines, antibodies, and diagnostic applications in parasitology.

### Author Summary

Antigen-antibody interactions are critical events in the humoral immune response, facilitating the recognition and neutralization of invasive parasites. BCEs, defined as surface-exposed clusters of amino acids recognized by B-cell receptors or antibodies, play a critical role in initiating a humoral immune response. This study focuses on the identification of parasite BCEs, which serve as promising targets for the development of vaccines, therapeutic antibodies, and diagnostic tools. To this end, we developed a deep learning model, termed deepBCE-Parasite, which was rigorously benchmarked against and integrated with traditional machine learning models. By leveraging state-of-the-art AI techniques, these models enable rapid and precise BCE identification directly from amino acid sequences, rendering it particularly suitable for large-scale epitope screening. As a proof-of-concept, we applied these AI-driven models to predict BCEs in *F. hepatica*, a globally distributed parasite responsible for fascioliasis, a neglected tropical disease. Utilizing available proteomics data of this trematode species, we identified peptides exhibiting high specificity for antibody binding. This work highlights the potential of AI in advancing epitope prediction within parasitology, providing a rapid, scalable, and cost-effective strategy for discovering immune targets.

## Introduction

Parasitic diseases, caused by both established and emerging infections, remain a significant global health challenge, particularly neglected tropical diseases (NTDs) associated with foodborne parasites [1]. These parasites often cause opportunistic infections in humans and livestock, imposing not only significant public health burdens on humans but also considerable and often underestimated economic losses in the livestock industry [2]. Notable examples of such NTDs include fascioliasis, caused by *Fasciola hepatica* and *Fasciola gigantica*, which pose significant economic and public health challenges globally [3,4]. Although conventional broad-spectrum antiparasitic drugs are effective in controlling or eradicating these parasites, several limitations and challenges hinder their long-term efficacy and safety.

Among these, the growing emergence of drug resistance represents as a major hurdle, driven by factors such as genomic evolution, improper drug administration, environmental factors, and limited availability of novel therapeutics [5,6]. Consequently, antibody-based therapies and vaccine development are increasingly being explored as promising alternatives for the prevention and control of parasitic infections.

Upon host invasion, parasite and its molecular components act as antigens that trigger host immune response. Some specific antibodies recognize BCEs, which are short amino acids sequences on antigenic proteins, thereby mediating humoral immunity against pathogens [7]. Nevertheless, the identification of BCE remains challenging due to the complex interactions between antibody variable regions and epitope sequences [8]. BCEs can be categorized as either conformational or linear epitopes [9]. Conformational epitopes comprise discontinuous amino acid residues that are spatially adjacent through protein folding, requiring structural modeling for precise prediction. In contrast, linear epitopes consist of continuous amino acid sequences that maintain their antigenicity even under denatured or partially unfolded conditions. Traditional in silico approaches with respect to the prediction of parasite BCEs predominantly depend on bioinformatics screening coupled with experimental immunogenicity validation [10–13]. Although computational methodologies have advanced significantly, the identification of BCEs using large-scale datasets has not been thoroughly investigated in parasitic immunology.

Machine learning and deep learning approaches have been extensively applied in the prediction of BCEs. For conformational epitopes, various algorithms, including Support Vector Machine (SVM), Adaptive Boosting (AdaBoost), Logistic Regression (LR), eXtreme Gradient Boosting (XGBoost), and pre-trained protein structure models (e.g., ESM-2), have demonstrated exceptional predictive accuracy [14–20]. Regarding linear BCEs, sequence-based approaches employing Recurrent Neural Networks (RNN), Language Models (LM), Extremely Randomized Tree (ERT), Gradient Boosting (GB), Random Forest (RF), and SVM have been extensively implemented [21–27]. In the field of parasitology, the post-genomic era has witnessed two significant developments: the completion of genomic sequencing projects for diverse protozoan and helminthic parasites [28,29], and considerable advancements in multi-omics technologies, particularly transcriptomics and proteomics [30–32]. These progressions have notably expanded parasite datasets, providing unprecedented opportunities to integrate AI models with multi-omics data, thereby enhancing prediction accuracy while reducing experimental costs.

In this study, we developed a Transformer-based deep learning model to predict linear BCEs in parasites. Trained on experimentally validated datasets from diverse parasite species, the model achieved an accuracy of 80.97% and an AUC of 0.9 on an independent test set. To benchmark its performance, we compared it with traditional machine learning approaches using 12 feature extraction techniques and four classifiers. Furthermore, we applied our models to proteomic data from the liver fluke *Fasciola hepatica*, predicting eight peptide sequences derived from the leucine aminopeptidase (LAP) protein. These peptides were synthesized and experimentally validated via dot-blot immunoassays, revealing that seven exhibited specific binding to antibodies in *F. hepatica*-positive ovine sera. Our findings demonstrate the efficacy of Transformer-based and machine learning models in predicting BCEs in parasites, providing valuable insights for the development of antibody-based therapeutics, peptide vaccines, and immunodiagnostic tools.

## Methods

### Dataset preparation of parasites

Although numerous prediction tools for BCEs have been developed (Table 1), none have been specifically trained on parasitic datasets, thus limiting their applicability to parasite-specific antigens. Moreover, the effective AI model training necessitates a well-curated dataset comprising both positive and negative samples. In this study, positive and negative samples of parasite BCEs were systematically collected from the Immune Epitope Database (IEDB) [33] and Bcipep database [34] (retrieved in April 2024). Initially, 8,128 positive BCE sequences were obtained. To mitigate redundancy and enhance dataset diversity for robust model training, sequence clustering was performed using CD-HIT v4.8.1 [35] with a

**Table 1. A summary of bioinformatics tools for conformational and linear BCE prediction.**

| Epitope | Predictor | Method | ACC | AUC | SE | SP | MCC | Reference |
|---|---|---|---|---|---|---|---|---|
| Conformational | CBTOPE | SVM | 0.87 | - | 0.83 | 0.90 | 0.73 | Ansari et al., [14] |
| | epitope3D | Adaboost | 0.70 | 0.78 | - | - | 0.55 | da Silva et al., [15] |
| | DiscoTope-3.0 | XGBoost | - | 0.81 | - | - | 0.52 | Høie et al., [16] |
| | SEMA 2.0 | ESM-2 | - | 0.78 | 0.70 | - | 0.23 | Ivanisenko et al., [17] |
| | EPSVR | SVR | - | 0.60 | - | - | - | Liang et al., [18] |
| | ElliPro | Thornton | 0.84 | 0.73 | 0.60 | 0.86 | - | Ponomarenko et al., [19] |
| | SEPPA 3.0 | LR | 0.67 | 0.75 | - | - | - | Zhou et al., [20] |
| Linear | LBCEPred | RF | 0.87 | 0.93 | 0.86 | 0.87 | 0.73 | Alghamdi et al., [21] |
| | BepiPred3 | LM | 0.69 | 0.76 | 0.70 | - | 0.31 | Clifford et al., [22] |
| | EpiDope | DNN | 0.77 | 0.67 | - | - | - | Collatz et al., [23] |
| | iBCE-EL | ERT, GB | 0.73 | 0.79 | 0.74 | 0.72 | 0.46 | Manavalan et al., [24] |
| | DeepLBCEPred | CNN, LSTM | 0.77 | - | 0.78 | 0.75 | 0.54 | Qi et al., [25] |
| | ABCpred | RNN | 0.66 | - | 0.67 | 0.65 | 0.32 | Saha et al., [26] |
| | SVMTriP | SVM | - | 0.70 | 0.80 | - | - | Yao et al., [27] |

similarity threshold of 0.8, yielding a refined dataset of 5,752 unique positive peptide sequences. The IEDB provides an experimentally validated negative dataset of parasite BCEs, which were extracted, deduplicated using the CD-HIT software, and numerically balanced to match the positive sample set. The final dataset was divided into training and test sets in an 80:20 ratio, ensuring rigorous model training and evaluation.

**Deep learning architecture**

**Embedding.** The model input consists of amino acid sequences, where each amino acid is represented as a dense vector through an embedding vector. This layer comprises two components: amino acid embedding and positional encoding.

(1) Amino acid embedding:

Each amino acid residue is mapped to a low-dimensional vector using a learned embedding matrix $E \in \mathbb{R}^{V \times d_k}$, where $V$ =24 (the vocabulary size, representing the 20 standard amino acids plus 4 special tokens) and $d_k$ =256 (the embedding dimension). The embedding matrix is optimized during training via backpropagation to capture semantic relationships between amino acids.

(2) Positional encoding:

To encode sequence order information, positional encoding is added to the amino acid embeddings. The encoding for position $p$ is computed as:

$$PE(p)_{2i} = \sin\left(\frac{p}{10000^{\frac{2i}{d_k}}}\right), \ PE(p)_{2i+1} = \cos\left(\frac{p}{10000^{\frac{2i}{d_k}}}\right)$$

Here, p denotes the position of the amino acid in the sequence, and $2i$, $2i+1$ represents even and odd dimensions of the positional encoding vector. The final embedding for each residue is the sum of its amino acid embedding and positional encoding:

$$X = \text{AminoAcidEmbedding}(a) + \text{PositionalEncoding}(p)$$

**Encoder-decoder architecture.**

(1) Multi-head attention mechanism:

An encoder-decoder architecture is built around a multi-head attention mechanism, which enables the model to focus on different parts of the input sequence simultaneously. This mechanism is implemented in both the encoder and decoder layers to capture complex dependencies within the sequences. For each attention head, queries ($Q$), keys ($K$), and values ($V$) are computed through learned linear transformations:

$$Q = W_Q \times X, \quad K = W_K \times X, \quad V = W_V \times X$$

Here, $X$ represents the input embeddings, and $W_Q$, $W_K$, and $W_V$ are learned weight matrices.

The attention scores are computed as:

$$\text{Attention}(Q, K, V) = \text{softmax}\left(\frac{QK^T}{\sqrt{d_k}}\right) V$$

where $d_k$=32 is the dimensionality of the keys. The scaling factor $\sqrt{d_k}$ ensures stable gradients during training. The multi-head attention mechanism applies this process across 8 parallel attention heads, allowing the model to capture diverse aspects of the input sequence. The outputs from all heads are concatenated and projected back into the model's dimensional space ($d_k$=256) using a learned linear transformation.

(2) Feed-forward network (FFN):

Both the encoder and decoder layers contain a FFN, which consists of two linear transformations with a ReLU activation function in between:

$$FFN(x) = max(0, xW_1 + b_1)W_2 + b_2$$

Here, $W_1 \in \mathbb{R}^{256 \times 2048}$ and $W_2 \in \mathbb{R}^{256 \times 2048}$ are learned weight matrices, and $b_1$, $b_2$ are bias terms. The FFN expands the hidden dimension to $d_{ff}$=2048 to enhance the model's capacity for nonlinear representation.

(3) Feature optimization block (FOB):

The FOB refines learned features using a 1D convolutional layer (kernel size 3, padding 1), followed by ReLU activation and dropout (p=0.3). The convolutional layer extracts local features, while dropout reduces overfitting. The output is projected back to the model's dimensional space ($d_k$=256) via a fully connected layer.

(4) Encoder layer:

The encoder layer consists of 2 sub-layers, namely multi-head attention and FFN. The input is first processed by the multi-head attention mechanism, followed by the FNN. Each sub-layer is accompanied by a residual connection and layer normalization:

$$EncoderOutput1 = Norm(FFN(\text{MultiHeadAttention}(x)))$$

In addition, we incorporate a FOB to further refine the learned features using convolutional layers and dropout.

$$EncoderOutput=FOB(EncoderOutput1)$$

(5) Decoder layer:

The decoder layer has a structure similar to the encoder but includes an additional encoder-decoder attention mechanism. This mechanism enables the decoder to focus on the encoder's output while processing the target sequence, integrating information from both the input and target sequences to generate predictions.

**Model output.** The final output from the decoder is passed through an adaptive average pooling layer, followed by a fully connected (FC) layer and a ReLU activation. Dropout is applied to prevent overfitting:

$$Output = ReLU(FC(Pool(\text{DecoderOutput})))$$

## Design of traditional machine learning models

Comparing the performance of different methods in the presence of a large number of handcrafted features is a complex task. To tackle this, we selected twenty representative statistical features as the foundation for our prediction and analysis. These features were derived from classical protein sequence encoding methods, including Amino Acid Composition (AAC), Adaptive Skip Dipeptide Composition (ASDC), Composition of k-Spaced Amino Acid Group Pairs (CKSAAGP), Composition of k-Spaced Amino Acid Pairs (CKSAAP), Dipeptide Deviation from Expected Mean (DDE), Di-Peptide Composition (DPC), Grouped Amino Acid Composition (GAAC), Grouped Di-Peptide Composition (GDPC), Grouped Tri-Peptide Composition (GTPC), Pseudo-Amino Acid Composition (PAAC), Quasi Sequence Order (QSO), and Sequence Order Coupling Number (SOCN). These features were extracted using the iLearnPlus tool [36].

To reduce noise in the feature matrix and negative effects on model performance, we employed the Light Gradient-Boosting Machine (LGBM) algorithm to rank all extracted features based on their importance. The proven capability of LGBM in feature selection makes it an ideal choice for enhancing model performance in high-dimensional tasks [37,38]. From this ranking, the top 200 features were selected. Subsequently, Recursive Feature Elimination (RFE) was applied to further fine-tune the selection, thereby improving both model performance and robustness.

For the classification task, we utilized four classic classification algorithms that are widely adopted in biological sequence analysis and prediction: SVM, RF, LGBM, and Gaussian Naïve Bayes (GNB). These algorithms were chosen due to their demonstrated effectiveness to handle complex, high-dimensional datasets, such as protein sequences, while maintaining robustness and generalizability. To implement and compare the performance of these features and classifiers, we used the scikit-learn (https://scikit-learn.org/stable/) library to execute machine learning algorithms.

## Model evaluation metrics

We assessed the performance of all models using four standard metrics: accuracy (ACC), sensitivity (SE), specificity (SP), and the Matthews correlation coefficient (MCC) [39–41].

$$ACC = \frac{TP + TN}{TP + FP + TN + FN} \times 100\%$$

$$SE = \frac{TP}{TP + FN} \times 100\%$$

$$SP = \frac{TN}{TN + FP} \times 100\%$$

$$MCC = \frac{(TP \times TN) - (FP \times FN)}{\sqrt{(TP + FP) \times (TN + FN) \times (TP + FN) \times (TN + FP)}}$$

In the aforementioned formulas, "True Positive" (TP) denotes the number of instances correctly classified as positive (i.e., accurately predicted BCEs), while "True Negative" (TN) represents the number of instances correctly classified as negative (i.e., accurately predicted non-BCEs). Conversely, "False Positive" (FP) indicates the number of negative instances erroneously classified as positive (i.e., non-BCEs misclassified as BCEs), and "False Negative" (FN) refers to the number of positive instances erroneously classified as negative (i.e., BCEs misclassified as non-BCEs). ACC indicates the proportion of correctly predicted instances, including both true positives and true negatives. SE measures the proportion of correctly predicted positives among all positive instances, while SP reflects the proportion of correctly predicted negatives among all negative instances. MCC provides a comprehensive assessment of model performance by considering all four prediction categories (TP, FP, TN, FN), offering a balanced assessment. Additionally, the Area Under the Curve (AUC) refers to the area under the Receiver Operating Characteristic (ROC) curve, which illustrates the model's ability to distinguish between positive and negative samples, with values closer to 1 indicating better classification performance.

## A case study for wet-lab evaluation

**Peptide identification in the liver fluke *Fasciola hepatica*.** To validate and evaluate the applicability of our models in parasitic research, we used *F. hepatica* as a case study. The raw proteomic sequencing data of *F. hepatica* were retrieved from the iProX database (https://www.iprox.cn/; Project ID: IPX0002165000). This dataset comprises mass spectrometry (MS)-based proteomic data from four developmental stages of *F. hepatica*: (1) Metacercaria, obtained from *Galba pervia* snails cultured under suitable conditions for 30–45 days, during which cercariae emerged and underwent cyst formation; (2) Juvenile fluke, isolated from the livers of sheep artificially infected with metacercariae for 28 days; (3) Immature fluke, recovered from the livers of sheep infected for 59 days; and (4) Adult fluke, extracted from the livers of sheep infected for 118 days [42]. The protein sequences of *F. hepatica* were downloaded from the UniProt database on October 12, 2024, using the keyword "*Fasciola hepatica*". Analysis of raw data was conducted using MaxQuant v2.6.5.0 [43], extracting key parameters from the proteinGroups file, including peptide counts, label-free quantification (LFQ) intensity values, sequence coverage, molecular weight, and FASTA headers. To facilitate cross-sample comparisons, LFQ intensity values were normalized using Z-score standardization.

**Model prediction of potential BCEs.** Based on MS-based proteomics data, we applied our models to predict potential linear BCEs of *F. hepatica*. To enhance prediction robustness, we conducted an intersection analysis across multiple models, retaining only epitopes consistently identified by different approaches. Furthermore, to elucidate the biological relevance of the predicted BCEs, we performed subcellular localization analysis of their source proteins using DeepLoc 2.0 [44] and assessed protein-protein interaction networks for key proteins via the STRING database [45].

**Synthesis, antigenicity, and experimental validation of BCEs via dot-blot immunoassays.** To experimentally validate the predicted BCEs, eight peptides derived from the LAP protein (Uniport ID: A0A345G0S4) were synthesized, alongside a human peptide sequence (RSRTPSLPTPPPTREP) used as a control. All peptides were chemically synthesized by Sangon Biotech Co., Ltd. (Shanghai, China), with the purity of the synthesized peptides confirmed to exceed 95% through high-performance liquid chromatography (HPLC) and MS analyses. Dot-blot immunoassays were performed on each nitrocellulose membrane (purchased from Beyotime Biotech Co., Ltd., Shanghai, China), with three peptides (10 μg of each) spotted per membrane. These membranes were air-dried, blocked with PBS containing 2% non-fat milk, and incubated overnight at 4°C with *F. hepatica*-positive and negative (control) ovine serum antibodies (diluted 1:400). Following washing, the membranes were incubated at room temperature for 1 hour with horseradish peroxidase (HRP)-conjugated rabbit anti-ovine IgG (H+L) (AS029, diluted 1:3000; ABclonal Biotech Co., Ltd., Wuhan, China). The binding signals were detected using the SuperPico ECL Master Mix (Vazyme Biotech Co., Ltd., Nanjing, China) and visualized with an enhanced chemiluminescence (ECL) detection system. Signal images were captured using a digital imaging platform. All peptides were tested in three independent experimental repetitions, and only peptides that consistently exhibited positive dot signals across all three replicates were confirmed as true BCEs specific to IgG antibodies.

## Results

### Design of the deep learning pipeline

We proposed a deep learning pipeline, named deepBCE-parasite, based on a custom Transformer architecture for predicting BCEs in parasite proteins. This model was trained on a balanced dataset containing 5,752 positive and 5,752 negative BCEs, with amino acid features encoded using amino acid embedding and positional encoding. The architecture employs a binary classification framework using an encoder-decoder structure (Fig 1A). The innovation is the integration of a multi-head attention mechanism within this encoder-decoder framework, which enables efficient feature extraction and context-aware sequence modeling (Fig 1B). Model hyperparameters include a vocabulary size of 24, an embedding dimension of 256, two encoder layers, eight attention heads, and a feedforward hidden layer dimension of 2,048. Detailed parameter settings are provided in S1 Table. This architecture offers a scalable and robust solution for epitope identification in complex parasite protein sequences.

The model was trained with a batch size of 512, utilizing 80% of the data for training and 20% for testing in each fold. Before training, as shown in S1 Fig, we conducted a comprehensive analysis of amino acid frequencies and peptide length distributions in both the training and test datasets. The results indicated that positive samples exhibited elevated frequencies of amino acids D, E, I, K, N, Q, and Y compared to negative samples, while amino acids A, G, H, L, P, R, S, T,

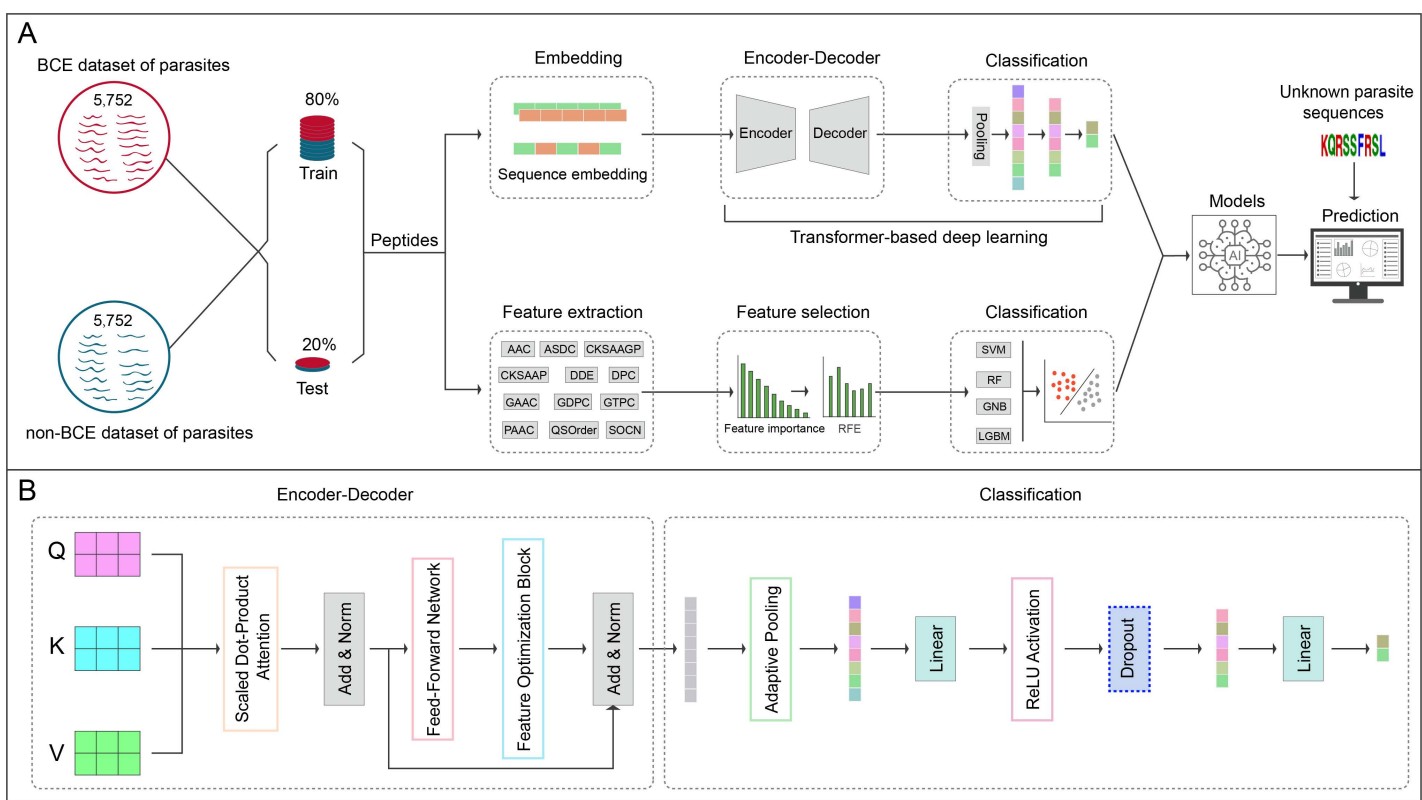

**Fig 1. Schematic representation of the study workflow.** (A) Overview of the proposed framework, highlighting a comparative architecture between the Transformer-based model (upper section) and the traditional machine learning model (lower section). (B) A detailed depiction of the Transformer-based model architecture, emphasizing its structure comprising 2 encoder layers and 8 attention heads. The model processes input amino acid sequences via an embedding layer, positional encoding, multi-head self-attention, and feature optimization modules, before classifying the sequences through a fully connected layer.

**Table 2. Performance of the Transformer-based deep learning model for predicting BCEs in parasites across training and independent test datasets.**

| Kernel | 10-Fold cross validation | | | | | Independent testing | | | | |
|---|---|---|---|---|---|---|---|---|---|---|
| | ACC (%) | AUC | SE (%) | SP (%) | MCC | ACC (%) | AUC | SE (%) | SP (%) | MCC |
| K=1 | 82.64 | 0.90 | 72.41 | 82.87 | 0.65 | 81.44 | 0.90 | 69.95 | 89.84 | 0.62 |
| K=2 | 81.95 | 0.89 | 71.93 | 81.97 | 0.64 | 80.78 | 0.90 | 70.31 | 88.45 | 0.60 |
| K=3 | 81.32 | 0.89 | 70.67 | 81.96 | 0.63 | 80.33 | 0.89 | 67.70 | 89.57 | 0.60 |
| K=4 | 82.38 | 0.90 | 72.36 | 82.40 | 0.65 | 80.93 | 0.90 | 69.60 | 89.23 | 0.61 |
| K=5 | 81.55 | 0.89 | 71.32 | 81.78 | 0.63 | 79.98 | 0.89 | 67.70 | 88.97 | 0.59 |
| K=6 | 82.05 | 0.89 | 71.61 | 82.50 | 0.64 | 81.59 | 0.90 | 69.95 | 90.10 | 0.62 |
| K=7 | 82.23 | 0.89 | 72.35 | 82.11 | 0.65 | 81.18 | 0.90 | 69.36 | 89.84 | 0.61 |
| K=8 | 81.74 | 0.89 | 71.98 | 81.51 | 0.64 | 81.23 | 0.90 | 72.21 | 87.84 | 0.61 |
| K=9 | 81.38 | 0.89 | 70.83 | 81.92 | 0.63 | 80.68 | 0.89 | 68.77 | 89.40 | 0.60 |
| K=10 | 81.88 | 0.89 | 71.90 | 81.86 | 0.64 | 81.59 | 0.90 | 70.90 | 89.40 | 0.62 |
| Mean | 81.91 | 0.89 | 71.74 | 82.09 | 0.64 | 80.97 | 0.90 | 69.64 | 89.26 | 0.61 |

ACC, accuracy; AUC, area under the curve; SE, sensitivity; SP, specificity; MCC, Matthews correlation coefficient

and V occurred less frequently in positive samples. Furthermore, analysis of length distribution demonstrated that positive samples predominantly contained peptides ranging from 10 to 15 amino acids in length, whereas negative samples were primarily 15 to 20 amino acids long. Upon completion of training, deepBCE-parasite achieved an average performance on the training set, with an ACC of 81.91%, an AUC of 0.89, a SE of 71.74%, a SP of 82.09%, and a MCC of 0.64. Rigorous evaluation via 10-fold cross-validation on the balanced BCE dataset yielded an average performance on the independent test set, with an ACC of 80.97%, an AUC of 0.90, a SE of 69.64%, a SP of 89.26%, and a MCC of 0.61. These results are summarized in Table 2.

### Selection of conventional machine learning models based on handcrafted features

Following the model establishment of the deep learning pipeline, we benchmarked its performance against traditional machine learning models to identify the most appropriate approaches for subsequent BCE screening. As shown in Fig 1A, this study also incorporated a traditional machine learning pipeline, which evaluated commonly used algorithms, including SVM, RF, LGBM, and GNB. To ensure a fair comparison, the same training and testing datasets were used across both deep learning and machine learning. A total of twelve handcrafted features were utilized in this study, including AAC, ASDC, CKSAAGP, CKSAAP, DDE, DPC, GAAC, GDPC, GTPC, PAAC, QSO, and SOCN, which were then applied to the four machine learning algorithms. Hyperparameters for the classifiers were optimized via Grid Search, with the detailed settings provided in S1 Table.

The performance results showed that features derived from the deep learning model consistently outperform those handcrafted features across all four algorithms, yielding a superior ROC curve (AUC=0.90) and highlighting the enhanced evaluation performance of deep learning over machine learning methods (S2 Fig). In addition to AUC, we evaluated several performance metrics, including SP, SE, and MCC. Notably, the RF classifier consistently achieved the highest classification performance compared to GNB, LGBM, and SVM, particularly when employing the CKSAAP, DDE, and DPC feature descriptors (S3 Fig). Specifically, for the independent test set, RF attained SP values of 81.67%, 84.71%, and 83.23% for CKSAAP, DDE, and DPC, respectively (Fig 2A), with corresponding SE values of 74.28%, 75.76%, and 75.07% (Fig 2B), and MCC values of 0.56, 0.61, and 0.58 (Fig 2C). Further analysis of the AUC values across the four classifiers showed the following results: for CKSAAP, the AUC values for GNB, LGBM, RF, and SVM were 0.71, 0.75, 0.78, and 0.73, respectively (Fig 2D); for DDE, the AUC values were 0.70, 0.76, 0.80, and 0.73 (Fig 2E); and for DPC, the

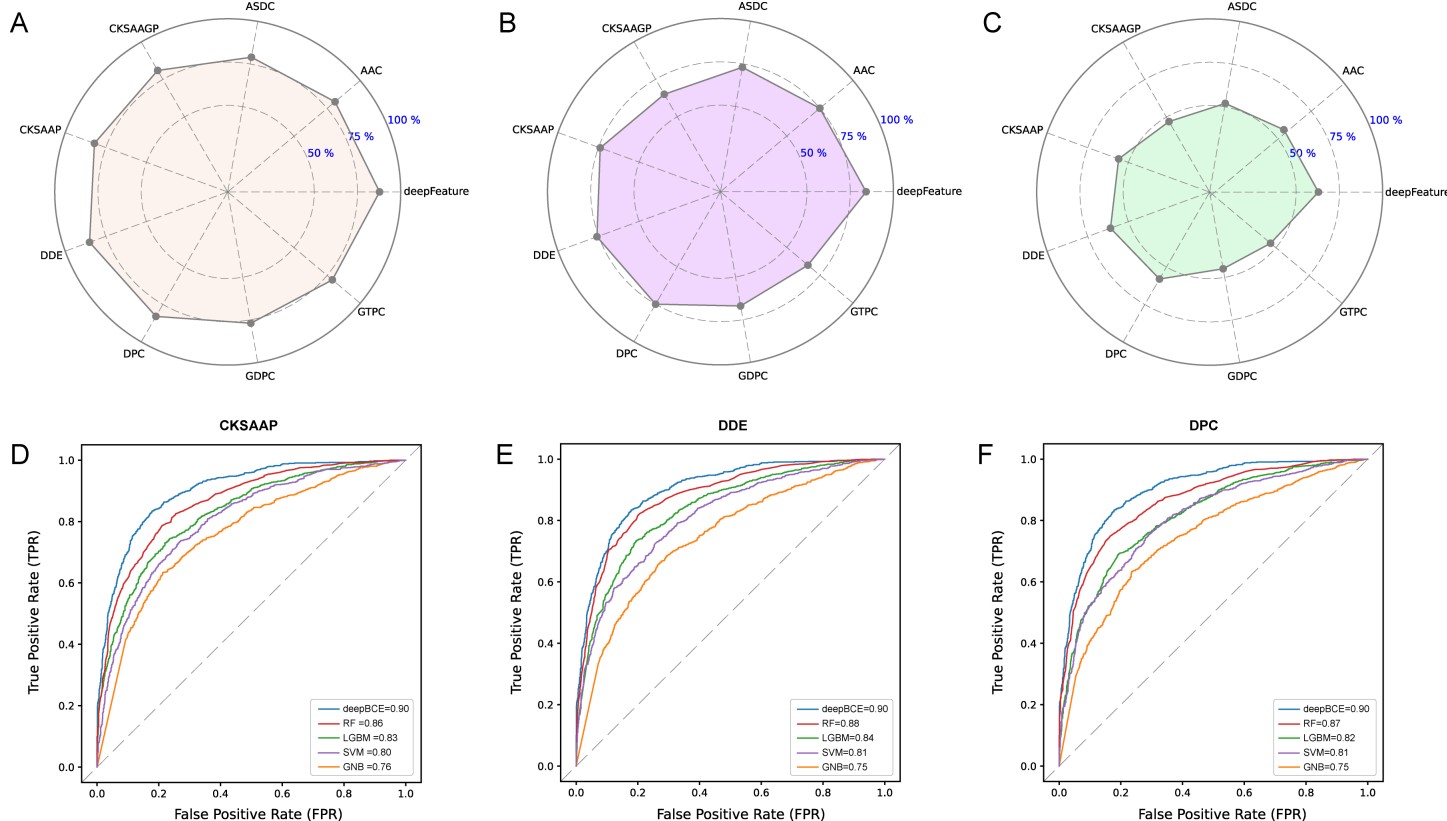

**Fig 2. Comparison of deep learning feature (deepFeature) with eight top traditional handcrafted features (AAC, ASDC, CKSAAGP, CKSAAP, DDE, DPC, GDPC, and GTPC).** (A-C) Radar plots showing SP, SE, and MCC values for each feature. (D-F) ROC curve comparing the deep learning model (deepBCE) with four traditional machine learning algorithms (GNB, LGBM, RF, and SVM) on CKSAAP, DDE, and DPC features. The x-axis represents False Positive Rate (FPR), and the y-axis represents True Positive Rate (TPR).

AUC values were 0.69, 0.74, 0.79, and 0.73 (Fig 2F). Additional performance metrics for machine learning methods are compiled in S2 Table. These findings further validate the superior performance of the RF classifier, confirming its selection as the optimal predictive models.

## Benchmarking custom and existing state-of-the-art models on dual test sets

To ensure a fair comparison between our models and existing linear BCE prediction models, we randomly partitioned two independent test sets, namely test1 and test2. As shown in Table 3 and Fig 3, deepBCE and BCE_RF are specialized models trained exclusively on experimentally validated datasets derived from the human and veterinary parasites, whereas EpiDope and BepiPred3 are general-purpose models trained on a wide range of BCE datasets. In the benchmark evaluation, BepiPred3 achieved remarkable SE with values of 99.46% and 98.31%, but its SP was notably low, reaching only 2.01% and 2.64%. This suggests that BepiPred3 is highly effective in identifying positive BCEs but performs inadequately in distinguishing negative BCEs. In contrast, although EpiDope surpassed BCE_RF in terms of AUC, it underperformed in other critical metrics, including SE and SP. Among our proposed models, deepBCE exhibited superior performance in discriminating positive samples and overall classification accuracy, while BCE_RF demonstrated slightly better capability in identifying negative samples and overall predictive performance. Overall, when evaluated in various metrics, the deepBCE and BCE_RF models, trained on parasite-specific BCE datasets, consistently surpassed EpiDope and BepiPred3, thereby highlighting their superior adaptability and effectiveness for parasite-related prediction tasks.

**Table 3. Performance comparison of our models against existing state-of-the-art models.**

| Model | Dataset | ACC (%) | AUC | SE (%) | SP (%) | MCC |
|---|---|---|---|---|---|---|
| deepBCE | test1 | 82.02 | 0.72 | 86.80 | 77.19 | 0.64 |
| BCE_RF | test1 | 81.20 | 0.55 | 79.20 | 85.22 | 0.65 |
| EpiDope | test1 | 59.95 | 0.64 | 66.73 | 53.10 | 0.20 |
| BepiPred3 | test1 | 50.95 | 0.57 | 99.46 | 2.01 | 0.07 |
| deepBCE | test2 | 81.34 | 0.73 | 86.47 | 76.27 | 0.63 |
| BCE_RF | test2 | 80.22 | 0.54 | 78.38 | 85.76 | 0.64 |
| EpiDope | test2 | 57.49 | 0.61 | 65.23 | 50.26 | 0.16 |
| BepiPred3 | test2 | 48.87 | 0.53 | 98.31 | 2.64 | 0.03 |

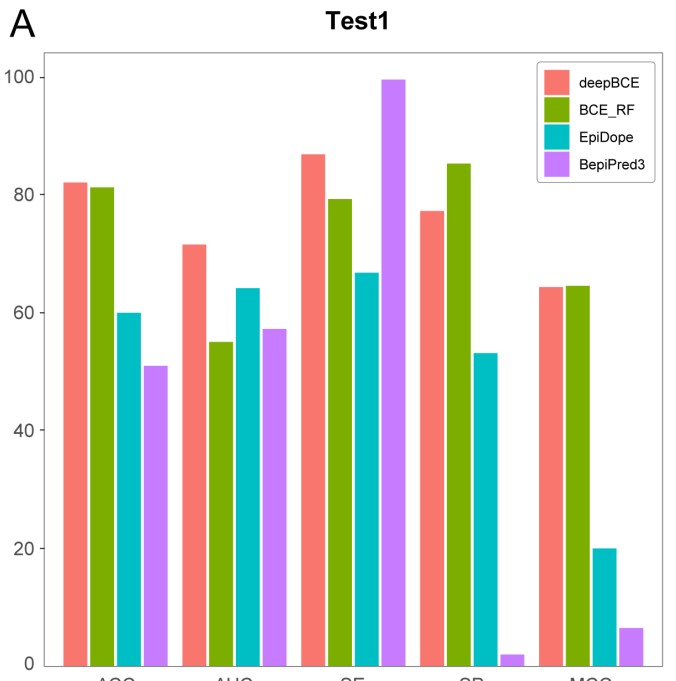
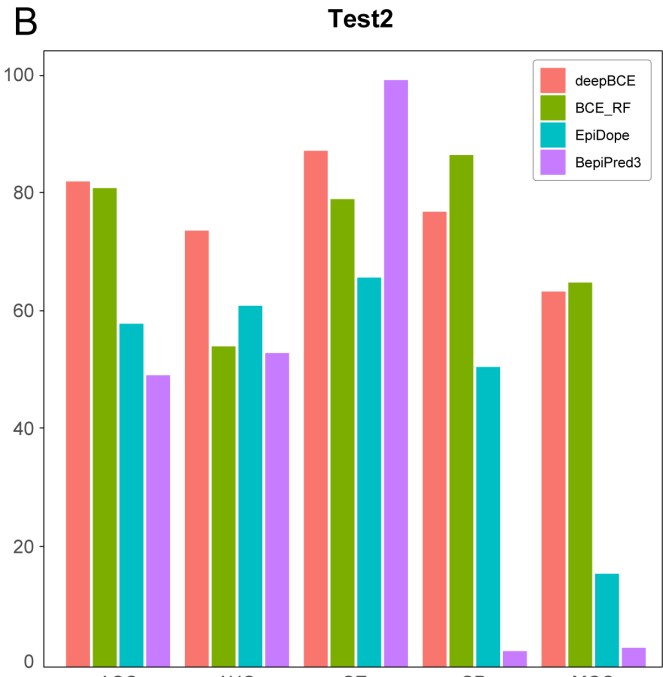

**Fig 3. Performance comparison between our models and existing models on benchmark datasets.** (A) Results on Test1 dataset. (B) Results on Test2 dataset.

## Proteomics-based bioinformatics analysis and prediction of BCEs in the liver fluke *Fasciola hepatica*

We evaluated the accuracy and utility of our predictive models for identifying BCEs using *F. hepatica* as a case study. Our models were applied to predict BCEs within *F. hepatica* proteins using proteomic data and peptide fragments. The complex life cycle of *F. hepatica* involves an intermediate snail host and a definitive mammalian host, typically cattle or sheep, although humans also act as definitive hosts (Fig 4A) [46]. Leveraging raw proteomic data, we characterized the proteome and expression profiles of *F. hepatica* across four developmental stages: metacercariae, juvenile fluke (28 days post-infection [dpi] in sheep), immature fluke (59 dpi), and adult fluke (118 dpi). Detailed protein identifications and corresponding expression profiles are provided in S3 Table. Based on these results, the following observations can be drawn.

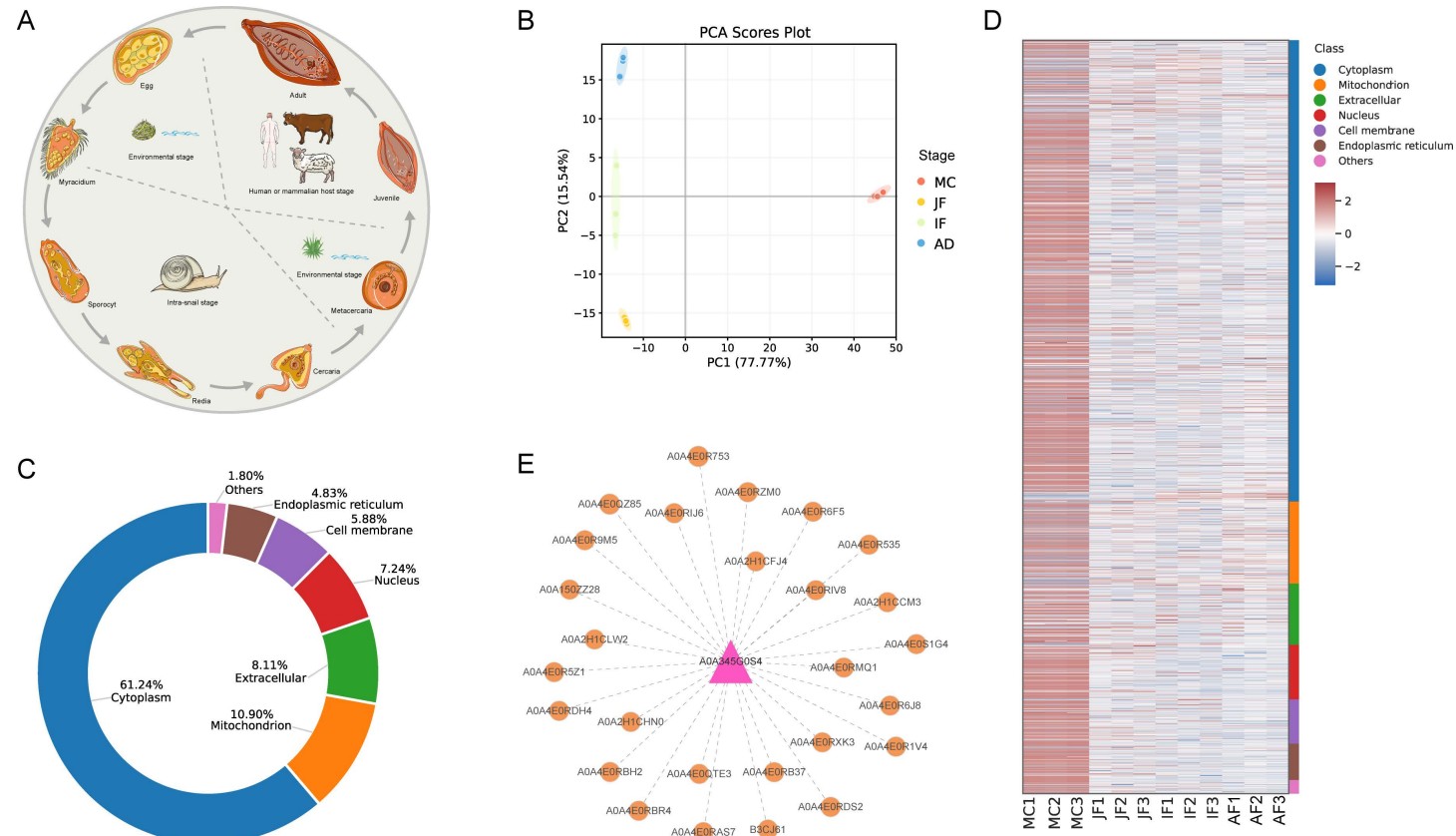

**Fig 4. Proteomic and bioinformatics analysis of *F. hepatica* across four developmental stages.** (A) The lifecycle of *Fasciola* spp., demonstrating the developmental stages within the intermediate snail host, environmental phases, and definitive mammalian/human host. The parasite image was adapted from Servier Medical Art (https://smart.servier.com/). (B) Principal component analysis (PCA) of protein expression profiles in metacercaria, juvenile fluke (28 dpi), immature fluke (58 dpi), and adult fluke (118 dpi). (C) Distribution of subcellular localization for proteins capable of generating B-cell epitopes (BCEs), highlighting their potential immunogenic properties. (D) Heatmap depicting protein expression dynamics across the four stages, with color-coded subcellular localization categories shown on the right. (E) Protein-protein interaction network of the leucine aminopeptidase (LAP) protein, generated using the STRING database, emphasizing key interactions and its potential as a vaccine candidate.

The most recent *F. hepatica* protein reference from the UniProt database was utilized to guide our MS analysis, resulting the identification of 2,165 proteins with a Q-value ≤ 0.01. PCA revealed pronounced differences in expression profiles across developmental stages, with the most substantial divergence observed between the metacercarial stage and the mammalian-host stages (Fig 4B). Additionally, the clustering of biological replicates across stages highlighted the high quality and reproducibility of the sequencing data, establishing a robust foundation for investigating the distribution of BCEs within parasitic proteins.

The deep learning model, combined with RF-based machine learning models, was employed to predict BCEs of *F. hepatica*. This integrated approach facilitates the identification of high-confidence BCEs in subsequent bioinformatics analysis, with the results summarized in S4 Table. To investigate the pre-cleavage subcellular localization of these source proteins, we systematically mapped their distribution across ten subcellular regions: nucleus, cytoplasm, extracellular space, mitochondria, cell membrane, endoplasmic reticulum, chloroplast, Golgi apparatus, lysosome/vacuole, and peroxisome. Our results indicate that the majority of BCE-source proteins were localized to the cytoplasm (61.24%), followed by mitochondria (10.90%), extracellular space (8.11%), nucleus (7.24%), cell membrane (5.88%), and endoplasmic reticulum (4.83%), with the remaining regions collectively accounting for 1.80% (Fig 4C). Moreover, heatmap clustering

demonstrated that proteins carrying BCEs showed the highest expression levels during the metacercariae (MC) stage, while significantly lower expression levels were observed in the mammalian stages (JF, IF, and AF) (Fig 4D).

Here, we selected four proteins: glutathione S-transferase (GST), leucine aminopeptidase (LAP), annexin (ANX), and *β*-actin, and conducted three-dimensional structure prediction using AlphaFold3 [47], followed by domain analysis using the InterPro web service [48]. As shown in Fig 5, the GST protein contains the GST_NTER and GST_C_Mu domains, yielding seven BCEs; the LAP protein harbors a Peptidase_M17 domain, resulting in eight BCEs; the ANX protein

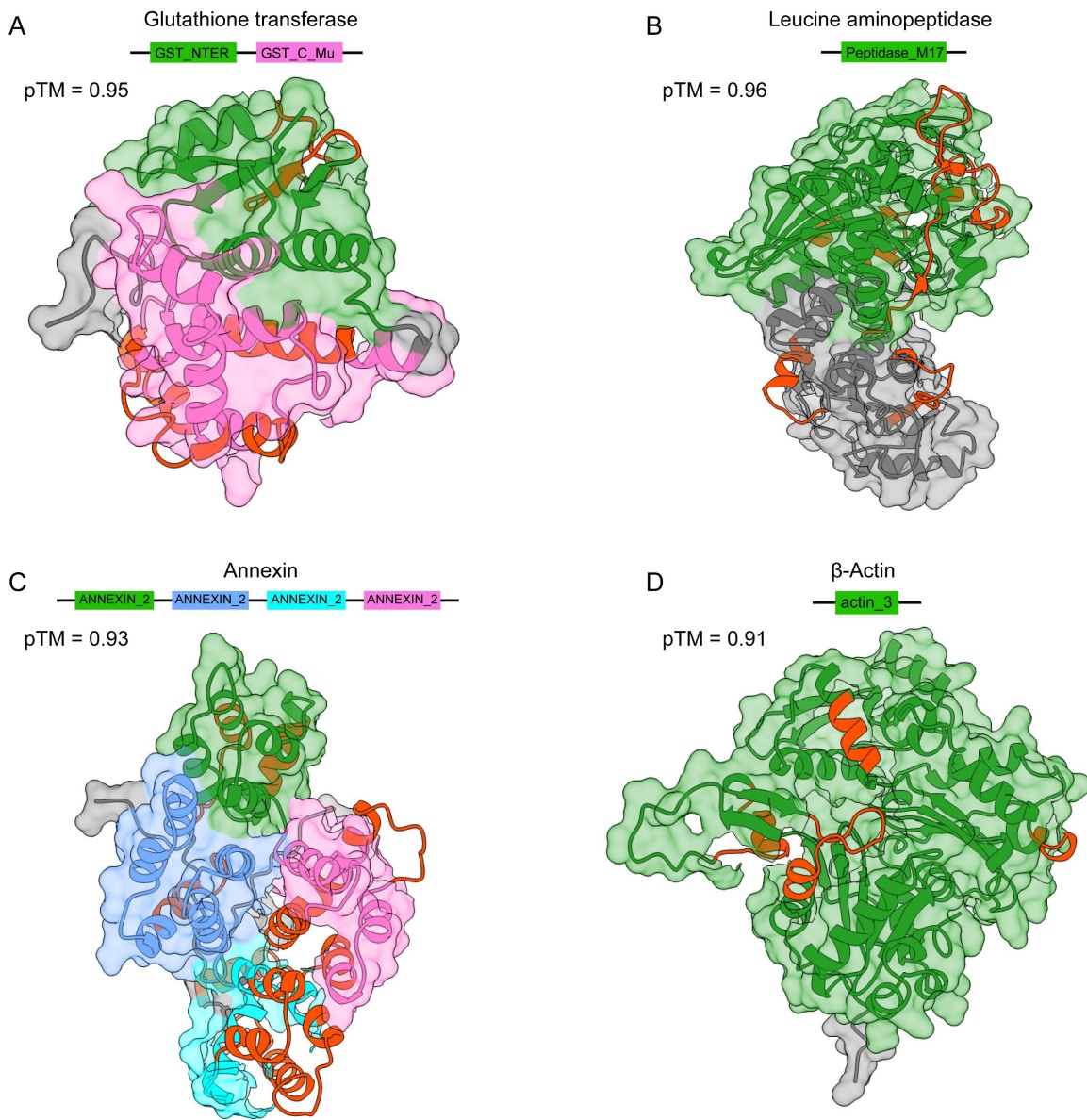

**Fig 5. Predicted 3D structural models of potential *F. hepatica* vaccine molecules, highlighting the spatial distribution of putative BCEs.** (A–D) Structural models for Glutathione transferase, leucine aminopeptidase, annexin, and *β*-actin, respectively. The predicted template modelling (pTM) score from AlphaFold3, which reflects prediction quality, is provided in the legend (higher value indicates greater model reliability). Predicted functional domains are color-coded, while non-domain regions are shown in gray.

includes four ANNEXIN_2 domains, producing ten BCEs; and the β-actin protein contains a single domain, generating four BCEs. Notably, in prior animal studies, the LAP protein demonstrated an 89% vaccine efficacy in sheep [49]. Protein interaction predictions further revealed that the LAP protein interacts with 27 high-confidence proteins, each of which also carries BCEs (Fig 4E), suggesting promising pathways for the exploration of *F. hepatica* immune targets.

### Dot-blot immunoassays

Structural domain prediction of the LAP protein identified the presence of a Peptidase_M17 domain, confirming its classification as a canonical member of the M17 family (Fig 5B). To assess its evolutionary conservation, a multiple sequence alignment was conducted using MAFFT [50], incorporating LAP protein sequences from diverse trematode species, including *Fasciola gigantica*, *Clonorchis sinensis*, *Opisthorchis felineus*, *Schistosoma japonicum*, and *Schistosoma mansoni*. The alignment indicated a high degree of conservation in predicted BCEs between *F. hepatica* and *F. gigantica*. In contrast, comparisons with other trematode species showed that only the sequence "DIGGSDPER" was fully conserved, while other predicted BCEs exhibited significant amino acid variability (Fig 6A).

To experimentally validate the predicted BCEs, eight peptides derived from the LAP protein was chemically synthesized and subjected to dot-blot immunoassays. Immunoreactivity analysis revealed that seven of the predicted BCEs (P2, P3, P4, P5, P6, P7, and P8) produced distinct dot signals in *F. hepatica*-positive ovine serum, while no dot signals were observed in synthetic human peptide controls and *F. hepatica*-negative ovine serum (Fig 6B and Table 4). In contrast, P1 failed to generate a detectable signal, suggesting a false-positive prediction by the model. Among the validated BCEs, P2, P4, P5, P6, and P8 were located within the predicted structural domain, whereas P3 and P7 were situated outside the domain. These findings demonstrate that the LAP protein exhibits multi-site antigenicity and has the potential to elicit host immune responses through distinct structural regions.

### Discussion

The identification of eukaryotic antigenic peptides derived from parasitic proteins previously relied on machine learning approaches that utilize handcrafted feature extraction, such as LGBM and stack-based models [51–53]. Although these approaches have demonstrated good prediction accuracy, their dependence on manual feature engineering often limits their ability to fully capture the complexity of antigenic peptide biology. In contrast, Transformer-based deep learning approaches have quickly emerged as a powerful alternative for protein sequence analysis, offering significant advantages over other deep learning methods, such as RNN-based models, which are constrained by sequential processing and struggle to model long-range dependencies [54–56]. By leveraging the Transformer architecture, the developed model effectively extracts hierarchical features that integrate amino acid properties and positional information through self-attention mechanisms. This approach not only captures both local and long-range residue interactions but also enables efficient parallel processing of large, variable-length sequences without the need for extensive preprocessing.

In this study, by integrating Transformer-based models with traditional machine learning models and leveraging publicly available *F. hepatica* proteomic data, potential BCEs were predicted and experimentally validated for those targeted by the LAP protein. Notably, seven of the eight synthesized peptides were confirmed as immunoreactive BCEs, demonstrating the high predictive accuracy and computational efficiency of our models in screening epitope candidates from large-scale MS-based datasets. The LAP protein was selected for its well-documented potential as a vaccine candidate that has been supported by extensive experimental evidence [57–61]. Our findings reveal the sequence characteristics and evolutionary relationships of LAP-driven BCEs, providing a bioinformatics foundation for peptide vaccine and diagnostic tool development. Additionally, proteins such as GST, ANX, and β-actin were identified as potential vaccine candidates based on their immunoprotective effects [62]. However, further experimental validation is required to confirm the antibody-binding efficacy of the BCEs derived from these candidate proteins.

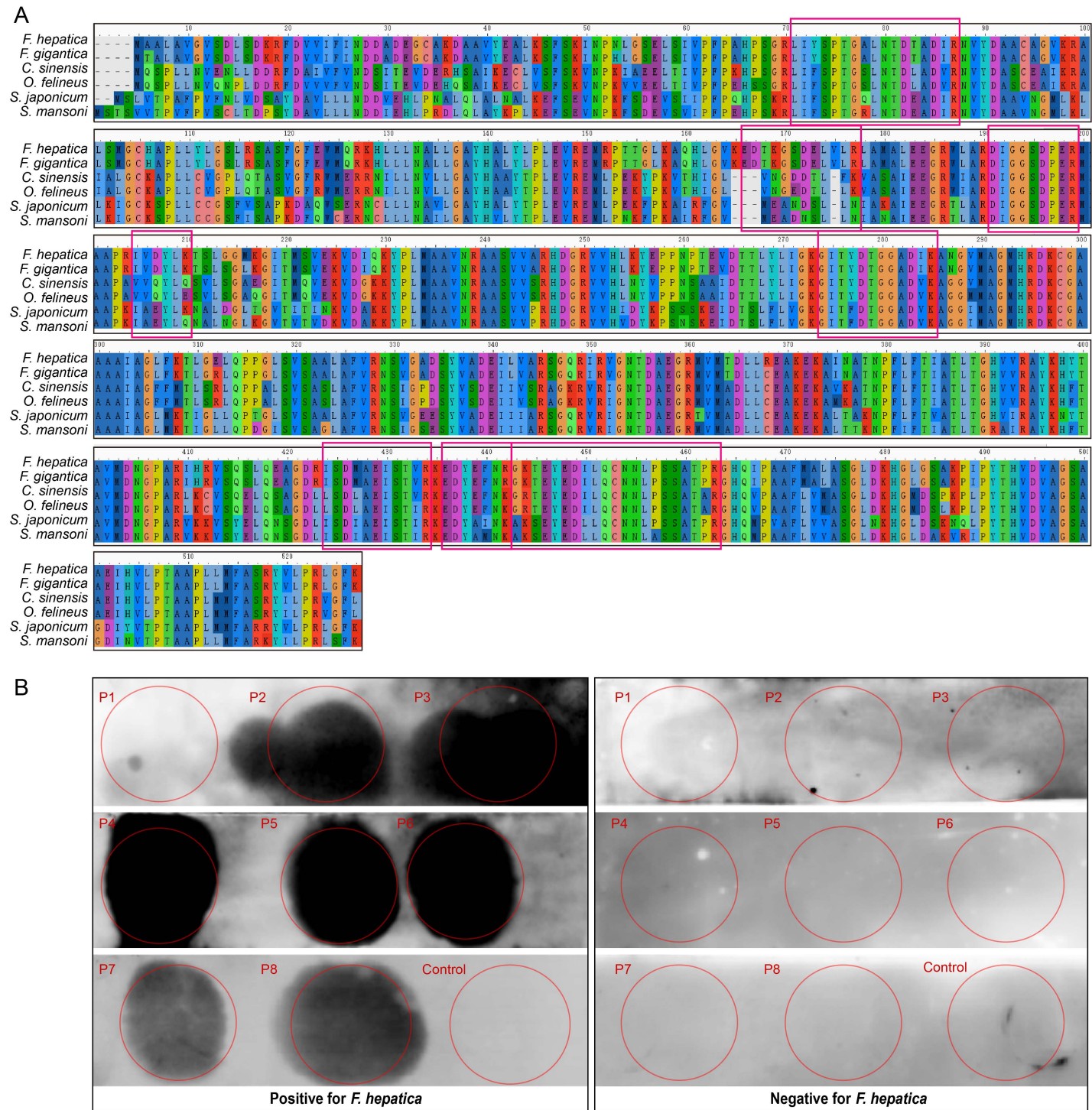

**Fig 6. Sequence alignment and experimental validation of the LAP protein and its putative BCEs.** (A) Multiple sequence alignment of *F. hepatica* LAP with five other trematode species. Predicted BCEs identified by our AI models are highlighted with red boxes. (B) Dot-blot immunoassay validation of predicted BCEs using serum from *F. hepatica*-positive animal, with *F. hepatica*-negative serum and a synthetic human peptide as controls.

**Table 4. Experimental validation of predicted BCEs derived from the LAP protein using dot-blot immunoassay.**

| Peptide name | Sequence | Position on LAP | Dot-blot |
| --- | --- | --- | --- |
| P1 | GKTEYEDILQCNNLPSSATPR | 439-459 | - |
| P2 | DIGGSDPER | 187-195 | + |
| P3 | LIYSPTGALNTDTADIR | 67-83 | + |
| P4 | ISDMAEISTVR | 420-430 | + |
| P5 | GITYDTGGADIK | 270-281 | + |
| P6 | EDYEFNR | 432-438 | + |
| P7 | EDTKGSDELVLR | 162-173 | + |
| P8 | IVDYLK | 201-206 | + |
| Control | RSRTPSLPTPPTREP | NA | - |

+, presence of a dot signal; -, absence of a dot signal

Importantly, our model is not restricted to the liver fluke *F. hepatica* but can be broadly applied to predict BCEs across a wide range of human and veterinary parasites, including both protozoan and helminth species. For example, in veterinary parasitology, this approach enables high-throughput screening of potential BCEs, allowing synthesized peptides to be combined with adjuvants and evaluated for their immunoprotective potential against pathogens [63]. Furthermore, integrating BCEs into diagnostic assays offers the potential to enhance the accuracy of infection detection, allowing for the differentiation between past and active infections, as well as the evaluation of vaccine efficacy. Although the current models demonstrate considerable predictive ability, further advancements are needed to improve the prediction of BCEs and to better capture complex structural interactions.

Our study only focuses on the development of linear BCE prediction models that leverage the linear sequence features of parasitic peptides to identify potential BCEs. Linear epitopes consist of contiguous amino acid sequences, which offer an advantage in model development due to their direct reliance on sequence features [64]. As a result, these epitopes are relatively straightforward to classify and experimentally validate. Consequently, linear epitope predictions are often prioritized in vaccine development and diagnostic design, particularly for rapid antigenic peptide identification [65–69]. However, this study does not address conformational epitope prediction, thus limiting its applicability in natural biological systems where BCEs typically adopt three-dimensional (3D) conformational structures. Conformational epitopes, formed by non-contiguous amino acid residues, depend on the overall protein folding and spatial exposure [70,71]. This dependence introduces additional challenges to epitope prediction, as it requires modeling complex structural interactions.

With advancements in AI-facilitated protein structure prediction, future studies on parasitic infections could benefit from protein models that integrate both linear and conformational epitope recognition. Open-source tools like AlphaFold, combined with structure determination techniques like X-ray crystallography and cryo-electron microscopy, would provide high-accuracy structures [72]. These advancements enable the identification of conformational epitopes, which are critical for simulating real-world antigen-antibody interactions. Future research should explore the development of multimodal predictive frameworks that integrate 3D structural information with sequence-based approach to improve the prediction of parasite-specific BCEs. Additionally, state-of-the-art deep learning architectures, such as 3D convolutional neural networks or graph neural networks [73,74], should be explored in the future to identify critical contact sites and surface-exposed features within protein structures.

## Conclusions

In this study, we developed a Transformer-based deep learning model alongside several traditional machine learning models for predicting BCEs in parasitic protein. Performance comparisons demonstrated that the Transformer-based model achieved superior predictive capability, with an independent test performance characterized by an ACC of 80.97%,

an AUC of 0.90, a SE of 69.64%, a SP of 89.26%, and a MCC of 0.61. Among the traditional machine learning models, the RF algorithm performed best, particularly when using the DDE feature descriptor, yielding an ACC of 80.23%, an AUC of 0.88, a SE of 75.76%, a SP of 84.71%, and a MCC of 0.61. In addition, this study leveraged proteomic data from *F. hepatica*, identifying seven BCEs through dot-blot immunoassay targeting the LAP protein, which specifically bind to *F. hepatica*-positive ovine serum antibodies, underscoring their potential as vaccine antigen candidates. The developed models show versatility, suitable for predicting BCEs in various parasitic pathogens, including protozoa and helminths. To enhance predictive accuracy and robustness in experimental validation, we recommend combining multiple models for BCE prediction in parasitic peptides.

## Supporting information

**S1 Fig. Amino acid frequency distribution and length statistics of peptides.** (A-B) Amino acid frequency in positive and negative samples from the training and testing datasets. (C-D) Peptide length distribution in positive and negative samples from the training and testing datasets.
(TIF)

**S2 Fig. Comparison of ROC curve between deep learning and 12 handcrafted features using four machine learning algorithms.**
(TIF)

**S3 Fig. Radar chart comparing the performance of deep learning features (deepFeature) with 12 handcrafted features across four machine learning algorithms.** The chart highlights their respective performance in SP, SE, and MCC metrics.
(TIF)

**S1 Table. Parameter settings for deep learning and machine learning models used in this study.**
(XLSX)

**S2 Table. Performance of conventional machine learning models with handcrafted features for B-cell epitope prediction in parasites.**
(XLSX)

**S3 Table. Bioinformatic analysis of proteomic features and expression profiles of *F. hepatica* across four developmental stages.**
(XLSX)

**S4 Table. The prediction results of B-cell epitope in *F. hepatica* using deep learning and machine learning models.**
(XLSX)

## Author contributions

**Conceptualization:** Kui Gu, Chun-Ren Wang.

**Data curation:** Rui-Si Hu, Muhammad Ehsan.

**Formal analysis:** Rui-Si Hu, Muhammad Ehsan.

**Funding acquisition:** Rui-Si Hu, Kui Gu, Chun-Ren Wang.

**Methodology:** Rui-Si Hu, Kui Gu, Sayed Haidar Abbas Raza, Chun-Ren Wang.

**Software:** Rui-Si Hu, Sayed Haidar Abbas Raza.

**Writing – original draft:** Rui-Si Hu.

**Writing – review & editing:** Kui Gu, Muhammad Ehsan, Sayed Haidar Abbas Raza, Chun-Ren Wang.

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
