## [Decision Letter · Decision Letter 0]

13 Jan 2025

Improved B-cell epitope identification in parasitic pathogens through a Transformer-driven model: Proof-of-concept in Fasciola hepatica

Dear Dr. Hu,

Thank you for submitting your manuscript to PLOS Neglected Tropical Diseases. After careful consideration, we feel that it has merit but does not fully meet PLOS Neglected Tropical Diseases's publication criteria as it currently stands. Therefore, we invite you to submit a revised version of the manuscript that addresses the points raised during the review process.

Please submit your revised manuscript within 60 days Mar 14 2025 11:59PM. If you will need more time than this to complete your revisions, please reply to this message or contact the journal office at plosntds@plos.org. Please include the following items when submitting your revised manuscript:

We look forward to receiving your revised manuscript.

Kind regards,

Aysegul Taylan Ozkan, M.D., Ph.D.,

Academic Editor

Jong-Yil Chai

Section Editor

Shaden Kamhawi

co-Editor-in-Chief

Paul Brindley

co-Editor-in-Chief

**Journal Requirements:**

At this stage, the following Authors/Authors require contributions: Rui-Si Hu, Kui Gu, Muhammad Ehsan, Sayed Haidar Abbas Raza, and Chun-Ren Wang. Please ensure that the full contributions of each author are acknowledged in the "Add/Edit/Remove Authors" section of our submission form.

2) Please ensure that all Table files have corresponding citations and legends within the manuscript. Currently, Table 2 in your submission file inventory does not have an in-text citation. Please include the in-text citation of the table.

3) Tables should not be uploaded as individual files. Please remove these files and include the Tables in your manuscript file as editable, cell-based objects. For more information about how to format tables, see our guidelines:

https://journals.plos.org/plosntds/s/tables 

Potential Copyright Issues:

i) Figures 1a, and 3a. Please confirm whether you drew the images / clip-art within the figure panels by hand. If you did not draw the images, please provide (a) a link to the source of the images or icons and their license / terms of use; or (b) written permission from the copyright holder to publish the images or icons under our CC BY 4.0 license. Alternatively, you may replace the images with open source alternatives. See these open source resources you may use to replace images / clip-art:

6) Your current Financial Disclosure states, "This work was supported by the National Natural Science Foundation of China [62202083]; the China Postdoctoral Science Foundation [2022M710618]; and the High-level Talent Research Start-up Project of Sichuan University of Arts and Science [WL085267]. ".

However, your funding information on the submission form indicates receiving one fund. Please ensure that the funders and grant numbers match between the Financial Disclosure field and the Funding Information tab in your submission form. Note that the funders must be provided in the same order in both places as well.                                                  . 

Please indicate by return email the full and correct funding information for your study and confirm the order in which funding contributions should appear. Please be sure to indicate whether the funders played any role in the study design, data collection and analysis, decision to publish, or preparation of the manuscript.

7) Please amend the label of Figure S3 in the online submission form as it is currently uploaded as Figure S2.

**Comments for the authors:**

**Please note that the reviews are uploaded as attachments.**

**Reviewers' Comments:**

Reviewer's Responses to Questions

**Key Review Criteria Required for Acceptance?**

**Methods**

-Are the objectives of the study clearly articulated with a clear testable hypothesis stated?

-Is the study design appropriate to address the stated objectives?

-Is the population clearly described and appropriate for the hypothesis being tested?

-Is the sample size sufficient to ensure adequate power to address the hypothesis being tested?

-Were correct statistical analysis used to support conclusions?

-Are there concerns about ethical or regulatory requirements being met?

Reviewer #1: Please see the more details annex file.

Reviewer #2: Authors have presented a well-executed study of predicting B-cell epitopes in F. hepatica using a deep learning model developed via transformer driven approach. Detailed comments are provided in the report.

Reviewer #3: The Methods section provides a comprehensive explanation of the Transformer architecture and methodology, but its technical density may overwhelm non-technical readers. For readers without a deep learning background, in order to improve accessibility and clarity, simplify the explanation of the Transformer architecture and consider including a flowchart or visual chart to illustrate key components such as attention layers and embeddings.

Also, please expand on the data preprocessing steps by describing how the dataset was balanced (e.g., oversampling) and detailing the processes for removing duplicates or noise.

Provide justification for the chosen hyperparameters, such as stating, "The choice of two encoder layers and eight attention heads was informed by prior studies and optimization experiments." Additionally, clarify the validation strategy by explicitly mentioning whether k-fold CV or an independent test set was used, and discuss measures taken to prevent overfitting. Finally, consider rephrasing overly technical sentences for broader readability. For instance, revise "The model utilized self-attention mechanisms to capture sequence dependencies" to "The model's attention mechanisms effectively identified patterns within the sequence data." These changes will make the section more engaging and accessible while maintaining its technical rigor.

Reviewer #4: This manuscript by Hu et al describes the development of a deep learning model, deepBCE-Parasite, which is applied in the identification of linear B-cell epitopes in parasitic pathogens, with a focus on Fasciola hepatica, a parasite of major health and economic importance. The authors have applied the Transformer architecture in designing a model that predicts BCEs from amino acid sequences using state-of-the-art self-attention mechanisms. In contrast, the deep learning approach significantly outperforms traditional machine learning methods such as Support Vector Machines and Random Forests with up to an accuracy of 81%, though robust performance across testing metrics is shown. The authors were able to use proteomic data from F. hepatica and predict eight BCEs derived from the leucine aminopeptidase protein, a known vaccine candidate, as a form of demonstration of its application. Experimental confirmation using dot-blot assays has been done on these, which shows the proof of concept that deep learning can help speed up BCE identification by reducing dependency on expensive and time-consuming experimental methods. It would be helpful in the development of vaccines, therapeutic antibody design, and diagnostics against neglected tropical diseases. In their paper, focusing on linear epitopes, the authors feel their future work needs to be directed towards conformational epitopes for better biological applicability. This manuscript is well written and I have a few suggestions for the improvement of the manuscript.

1. While the authors tested the model on Fasciola hepatica, its utility across other parasitic species was not thoroughly explored. The authors could test the model on other protozoan and helminthic parasites to demonstrate its generalizability and robustness in predicting BCEs for a wider range of pathogens.

2. Experimental validation was limited to eight peptides from a single protein. This small sample size may not fully reflect the model's predictive accuracy or its applicability to diverse proteomes. The authors should acknowledge this limitation.

3. While the model outperformed traditional machine learning approaches, it was not compared to state-of-the-art epitope prediction tools like DiscoTope or BepiPred-3.0. The authors could benchmark deepBCE-Parasite against other advanced models to establish its competitive edge and address any potential limitations in its methodology.

4. The authors should clearly state the limitations of their study.

Reviewer #5: This article comprehensively addresses the development of both deep learning and traditional machine learning models for predicting linear B cell epitopes (BCEs) in parasitic organisms, focusing on protozoa and helminths. The manuscript successfully demonstrates the benefits of Transformer-based architectures (in particular, its encoder-decoder approach with multi-head attention) over traditional methods, while providing an insightful case study on Fasciola hepatica. The validation of leucine aminopeptidase (LAP)-derived epitopes in the lab lends additional credence to the computational predictions.

1.Some details—particularly hyperparameter choices and rationale—could be expanded for reproducibility. For instance, justify why certain embedding dimensions or numbers of heads were selected.

Comparison Across Multiple Models

2. Add more discussion on how these findings could accelerate or integrate into wider vaccine and diagnostic test pipelines in veterinary or human parasitology.

3. Increase the clarity around experimental methods (e.g., how peptides were chosen for validation and how many repetitions were performed). If possible, include quantification (intensity measurements) in a table or figure to show how signal intensities compare across peptides.

4. Provide a clearer roadmap of how the proposed pipeline can be extended to 3D data and conformational epitopes in future work. A brief discussion on potential synergy with AlphaFold or related structural modeling tools would be beneficial.

**Results**

-Does the analysis presented match the analysis plan?

-Are the results clearly and completely presented?

-Are the figures (Tables, Images) of sufficient quality for clarity?

Reviewer #1: Please see the more details annex file.

Reviewer #2: Work is well-executed and explained but figures of the main manuscript lack clarity, tremendously. High resolution images are needed.

Reviewer #3: In the Results section, while performance metrics such as AUC and MCC are provided, their significance in the context of vaccine design is not clearly articulated.

For example, why does achieving an AUC of 0.90 matter, and how does it advance the field? Similarly, although the figures and tables are informative, do the captions provide enough detail to ensure clarity and proper interpretation for the reader?

To enhance this section, qualitative examples would help demonstrate the model's real-world utility. For instance, highlighting specific epitopes that the Transformer model successfully identified but traditional methods missed would strengthen the narrative. A brief case study showing the model’s practical application could further illustrate its value.

Moreover, explaining why the Transformer outperforms traditional methods like SVM or Random Forest would add depth to the discussion—what aspects of the model contribute to its superior performance?

Finally, acknowledging any limitations of the model, such as specific types of epitopes it struggles to predict, or potential dataset biases, would provide a more balanced perspective.

Other comments for experiment validation

Why did the authors choose Fasciola hepatica as the research subject to validate the effectiveness of the model? Also, the authors highlighted four proteins, namely Glutathione transferase, Leucine aminopeptidase, Annexin, and β-Actin, to present the results. However, in the context of Fasciola hepatica research, these proteins have not demonstrated particularly ideal immune protection in livestock animals, even though they may exhibit some immunogenic effects. Additionally, can the deep learning and machine learning models developed in this study be applied to research on other parasites? And explain how this method could be applied to other parasitic pathogens or even other classes of pathogens (e.g., viral or bacterial epitopes)?

Reviewer #4: Yes, the analysis aligns with the outlined plan. The manuscript describes the development and validation of the deepBCE-Parasite model.

Yes, the results are generally well-organized and clearly presented

Yes the figures are of good quality.

Reviewer #5: Reiterate or cross-reference the original analysis plan in the Results section to show how each step (e.g., cross-validation, feature selection, benchmarking) was carried out exactly as proposed.

**Conclusions**

-Are the conclusions supported by the data presented?

-Are the limitations of analysis clearly described?

-Do the authors discuss how these data can be helpful to advance our understanding of the topic under study?

-Is public health relevance addressed?

Reviewer #1: Please see the more details annex file.

Reviewer #2: Authors have addressed the limitations of the study at the current stage.

Reviewer #3: The conclusions are well supported by the data presented.

Reviewer #4: Yes, the conclusions are supported by the data.

The authors acknowledge some limitations, such as the model's focus on linear epitopes and the lack of conformational epitope prediction, which reduces its applicability to real-world antigen-antibody interactions. However, they do not fully discuss other potential shortcomings, such as dataset bias, limited validation across diverse parasites, or the relatively small scope of experimental validation.

Yes. The authors emphasize the significance of their model in advancing epitope-based vaccine design, antibody therapeutics, and diagnostics for parasitic diseases.

Yes. The authors address the public health implications of their work, noting that Fasciola hepatica is a globally significant zoonotic parasite and emphasizing the potential of the model to aid in vaccine development and diagnostic tools for neglected tropical diseases.

Reviewer #5: Strengthen the connection to public health by explicitly mentioning how more accurate BCE prediction can reduce infection rates, improve diagnostics, or inform surveillance programs.

**Editorial and Data Presentation Modifications?**

Reviewer #1: Please see the more details annex file.

Reviewer #2: Authors have provided the background of current study and highlighted the limitations/challenges followed by desired future actions. The work can be accepted after addressing some minor concerns:

• Highlight the importance of LGBM algorithm along with supporting references, for better understanding of the readers.

• Authors should also discuss other limitations/challenges (in addition to rising drug-resistance) associated with the use of ‘conventional antiparasitic broad-spectrum drugs’, in the current scenario.

• Authors should highlight the possible advantages of incorporating artificial intelligence tools in such practices, to support or compliment experimental studies.

Reviewer #3: Hu et al., presents an innovative approach to B-cell epitope prediction using a Transformer-based deep learning model, specifically focusing on parasitic pathogens. While the study demonstrates strong technical merit and relevance to vaccine design, several issues require attention to improve the overall clarity, methodological rigor, and presentation. The overall writing and logic are smooth, but the content and spelling throughout the text need careful checking. Some information need explained and enhance clarified. I wish to recommend Minor Revision.

Reviewer #4: Minor Revision

Reviewer #5: (No Response)

**Summary and General Comments**

Reviewer #1: Please see the more details annex file.

Reviewer #2: Manuscript is very well written.

Reviewer #3: Hu et al., presents an innovative approach to B-cell epitope prediction using a Transformer-based deep learning model, specifically focusing on parasitic pathogens. While the study demonstrates strong technical merit and relevance to vaccine design, several issues require attention to improve the overall clarity, methodological rigor, and presentation. Below, I provide detailed comments and suggestions for each section of the manuscript.

In the abstract part, the author provides a concise overview of the study, including the problem addressed, methodology, and results. However, it is overly technical and does not clearly articulate the novelty and significance of the research. For example, while metrics such as AUC are mentioned, their relevance is not explained. Additionally, the concluding statement lacks an impact-oriented summary of how the findings could contribute to vaccine design. I have some suggestions that require authors to clarify and revise them:

First, please simplify the technical jargon to make it accessible to a broader audience. For instance, it perhaps more suitable for replacing "Transformer-driven model leveraging self-attention mechanisms" with "a novel deep learning model that improves prediction accuracy using advanced attention techniques.

Second, clearly state the novelty of the approach, such as its ability to outperform traditional machine learning models in specific metrics.

Lastly, can add a sentence on the broader implications of the study, e.g., "This study provides a robust tool for identifying B-cell epitopes, which could significantly enhance the development of vaccines for parasitic diseases."

The introduction provides a strong foundation on epitope prediction and underscores the importance of B-cell epitopes in vaccine development. However, the narrative is disrupted by an excessive focus on datasets and existing tools early on, and while the limitations of traditional methods are acknowledged, the specific research gap that this study addresses remains unclear. Reorganizing the introduction could significantly improve its clarity and flow. Thus, it is recommended to start with the significance of B-cell epitopes in vaccine design, followed by the challenges of existing prediction methods, such as limited accuracy and reliance on hand-crafted features. Moreover, clearly articulate the research gap and objective by stating, "To address these challenges, we developed a Transformer-based model to improve the accuracy and applicability of B-cell epitope prediction." Avoid redundancy, such as repeating the mention of the IEDB database, and briefly highlight the relevance of parasitic diseases like Fasciola hepatica to contextualize the study's focus. This approach will create a more engaging and logically structured introduction.

The Methods section provides a comprehensive explanation of the Transformer architecture and methodology, but its technical density may overwhelm non-technical readers. For readers without a deep learning background, in order to improve accessibility and clarity, simplify the explanation of the Transformer architecture and consider including a flowchart or visual chart to illustrate key components such as attention layers and embeddings.

Also, please expand on the data preprocessing steps by describing how the dataset was balanced (e.g., oversampling) and detailing the processes for removing duplicates or noise.

Provide justification for the chosen hyperparameters, such as stating, "The choice of two encoder layers and eight attention heads was informed by prior studies and optimization experiments." Additionally, clarify the validation strategy by explicitly mentioning whether k-fold CV or an independent test set was used, and discuss measures taken to prevent overfitting. Finally, consider rephrasing overly technical sentences for broader readability. For instance, revise "The model utilized self-attention mechanisms to capture sequence dependencies" to "The model's attention mechanisms effectively identified patterns within the sequence data." These changes will make the section more engaging and accessible while maintaining its technical rigor.

In the Results section, while performance metrics such as AUC and MCC are provided, their significance in the context of vaccine design is not clearly articulated.

For example, why does achieving an AUC of 0.90 matter, and how does it advance the field? Similarly, although the figures and tables are informative, do the captions provide enough detail to ensure clarity and proper interpretation for the reader?

To enhance this section, qualitative examples would help demonstrate the model's real-world utility. For instance, highlighting specific epitopes that the Transformer model successfully identified but traditional methods missed would strengthen the narrative. A brief case study showing the model’s practical application could further illustrate its value.

Moreover, explaining why the Transformer outperforms traditional methods like SVM or Random Forest would add depth to the discussion—what aspects of the model contribute to its superior performance?

Finally, acknowledging any limitations of the model, such as specific types of epitopes it struggles to predict, or potential dataset biases, would provide a more balanced perspective.

Other comments for experiment validation

Why did the authors choose Fasciola hepatica as the research subject to validate the effectiveness of the model? Also, the authors highlighted four proteins, namely Glutathione transferase, Leucine aminopeptidase, Annexin, and β-Actin, to present the results. However, in the context of Fasciola hepatica research, these proteins have not demonstrated particularly ideal immune protection in livestock animals, even though they may exhibit some immunogenic effects. Additionally, can the deep learning and machine learning models developed in this study be applied to research on other parasites? And explain how this method could be applied to other parasitic pathogens or even other classes of pathogens (e.g., viral or bacterial epitopes)?

(6)The overall writing and logic are smooth, but the content and spelling throughout the text need careful checking. For example, on line 241, the abbreviation for Gaussian Naive Bayes should be GNB.

Reviewer #4: Minor Revision

Reviewer #5: Overall, the study’s design, methodology, and results are presented cohesively. Once the minor revisions—particularly regarding clarity on experimental methods, dataset annotation, and limitations—are addressed, the manuscript will be suitable for publication. The revised paper would serve as a valuable resource for researchers working on vaccine design and diagnostic test development in parasitology and related fields.

PLOS authors have the option to publish the peer review history of their article (what does this mean? ). If published, this will include your full peer review and any attached files.

**Do you want your identity to be public for this peer review?** For information about this choice, including consent withdrawal, please see our Privacy Policy .

Reviewer #1: No

Reviewer #2: No

Reviewer #3: No

Reviewer #4: No

Reviewer #5: **Yes: ** Simna Saraswathi Prasannakumari

**Figure resubmission:**

**Reproducibility:**



---

## [Decision Letter · Decision Letter 1]

13 Mar 2025

Dear Dr Hu,

We are pleased to inform you that your manuscript 'Transformer-based deep learning enables improved B-cell epitope prediction in parasitic pathogens: A proof-of-concept study on Fasciola hepatica' has been provisionally accepted for publication in PLOS Neglected Tropical Diseases.

Best regards,

Aysegul Taylan Ozkan, M.D., Ph.D.,

Academic Editor

Jong-Yil Chai

Section Editor

Shaden Kamhawi

co-Editor-in-Chief

Paul Brindley

co-Editor-in-Chief

Reviewer's Responses to Questions

**Key Review Criteria Required for Acceptance?**

**Methods**

-Are the objectives of the study clearly articulated with a clear testable hypothesis stated?

-Is the study design appropriate to address the stated objectives?

-Is the population clearly described and appropriate for the hypothesis being tested?

-Is the sample size sufficient to ensure adequate power to address the hypothesis being tested?

-Were correct statistical analysis used to support conclusions?

-Are there concerns about ethical or regulatory requirements being met?

Reviewer #2: The manuscript is acceptable in the current form comprises of all the above aspects.

Reviewer #3: YES

Reviewer #4: The authors responded satisfactorily to all the comments and made the required changes in the manuscript.

**Results**

-Does the analysis presented match the analysis plan?

-Are the results clearly and completely presented?

-Are the figures (Tables, Images) of sufficient quality for clarity?

Reviewer #2: Yes.

Reviewer #3: YES

Reviewer #4: The authors responded satisfactorily to all the comments and made the required changes in the manuscript.

**Conclusions**

-Are the conclusions supported by the data presented?

-Are the limitations of analysis clearly described?

-Do the authors discuss how these data can be helpful to advance our understanding of the topic under study?

-Is public health relevance addressed?

Reviewer #2: Yes.

Reviewer #3: YES

Reviewer #4: The authors responded satisfactorily to all the comments and made the required changes in the manuscript.

**Editorial and Data Presentation Modifications?**

Reviewer #2: Accept.

Reviewer #3: Accepted

Reviewer #4: The authors responded satisfactorily to all the comments and made the required changes in the manuscript.

**Summary and General Comments**

Reviewer #2: The manuscript has been significantly improved after the revision and can be accepted in the current form.

Reviewer #3: Accepted

Reviewer #4: The authors responded satisfactorily to all the comments and made the required changes in the manuscript.

PLOS authors have the option to publish the peer review history of their article (what does this mean? ). If published, this will include your full peer review and any attached files.

**Do you want your identity to be public for this peer review?** For information about this choice, including consent withdrawal, please see our Privacy Policy .

Reviewer #2: No

Reviewer #3: No

Reviewer #4: No